# Soil Management Systems to Overcome Multiple Constraints for Dryland Crops on Deep Sands in a Water Limited Environment on the South Coast of Western Australia

**David J. M. Hall [1],\*, Stephen L. Davies [2], Richard W. Bell [3]** 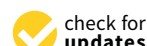 **and Tom J. Edwards [1]**

1   Department of Primary Industries and Regional Development, PMB 50, Melijinup Rd, Esperance 6450, Australia; tom.edwards@dpird.wa.gov.au
2   Department of Primary Industries and Regional Development, 20 Gregory Street, Geraldton 6530, Australia; stephen.davies@dpird.wa.gov.au
3   Agriculture Discipline, College of Science Health Engineering and Education, Murdoch University, South St, Murdoch 6150, Australia; R.Bell@murdoch.edu.au
\*   Correspondence: david.hall@dpird.wa.gov.au

**Abstract:** Deep sands on the south coast sandplain of Western Australia (WA) have multiple soil constraints including water repellence, high soil strength, low nutrient levels and subsoil acidity. The aim of the study was to test contrasting methods of managing water repellence and to assess their impacts on one or more soil constraints to crop production. These methods included seeding tyne design (knife point, winged points, paired row), soil wetting agent addition, strategic inversion tillage (rotary spading, mouldboard ploughing to 0.35 m) and clay-rich subsoil addition (170 t ha$^{-1}$ with incorporation by spading to 0.20 or 0.35 m). Limesand (2 t ha$^{-1}$) was applied as a split plot treatment prior to tillage. Cumulative crop yields were increased by 2.1–2.6 t ha$^{-1}$ over five years by the strategic deep tillage and clay application treatments compared to the control. Water repellence was reduced by the inversion ploughing and subsoil clay addition treatments only. The effect of water repellence on crop establishment was expressed only in low rainfall years (Decile < 4) and mitigated by the paired row, wetting agent, spader and clay-amended treatments. In all years, plant numbers were adequate to achieve yield potential regardless of treatment. Soil K and plant tissue K and B were increased where clay had been applied. Inversion tillage reduced soil pH, organic carbon (OC) and macro nutrients in the 0–0.1 m layer although in most years there was no significant decline in plant tissue macro nutrient levels. Soil strength was reduced as a result of the inversion tillage to a depth of 0.35 m. However, the alleviation of soil strength and the crop yield responses diminished with time due to re-compaction. No crop response to the applied lime was found over five years at this site since the soil pHCaCl$_2$ exceeded 4.7 within the root zone. In terms of soil constraints, we conclude that compaction was the dominant constraint at this site followed by water repellence and K deficiency.

**Keywords:** deep sands; strategic tillage; soil water repellence; soil compaction; inversion ploughing; clay addition; wetting agent; seeding tine design; limesand

## 1. Introduction

Approximately 900,000 km$^2$ or 5% of soils used for crop production globally have a deep sand profile [1]. The proportion of cropping soils with sand subsoil is greatest in Africa and Australia-Oceania. A characteristic of deep sands is that they present multiple constraints to crop production, mostly related to their limited capacity to store and supply water and nutrients to roots. The Deep sands (>0.3 m) cover

approximately 40% (7 Million ha) of the Western Australia (WA) wheatbelt and are predominately used for dryland crop and volunteer pasture production. These soils are subject to a number of constraints to crop production [2] which relate to their parent material, texture and soil forming processes. The south coast sandplain soils are derived from Precambrian granites which have undergone marine incursions during the Eocene and the deposition of aeolian sand sheets during the Quaternary epoch [3]. The multiple constraints on these sands include acidity (pHCaCl$_2$ < 5.5), low cation exchange (<4 cmol (+) kg$^{-1}$), low pH buffering capacity and inherently low levels of macro and micro nutrients [4]. Given their aeolian genesis, the sandplain soils contain a sufficiently wide particle size distribution within the sand fractions to allow dense packing [5,6]. The ability to pack densely is exacerbated through natural and induced compactive forces including wetting and drying cycles [7] and trafficking by machinery [8,9]. Water repellence is common in sandplain soils of WA due to the topsoils having a low proportion of clay (<5%) combined with naturally occurring organic waxes and polymers [10]. Consequently, the sandplain soils on the south coast of WA are prone to multiple soil constraints for dryland crop production including compaction, water repellence, nutrient deficiencies and subsoil acidity [2,11,12].

Where there are multiple limitations, the process of soil amelioration often seeks to sequentially overcome constraints starting with the most yield-limiting one. Pale deep sands ([13], Albic Arenosols) and sandy earths ([13], Ferralic Cambisols) in WA are considered to be highly responsive soils to deep tillage [14]. Mean increases in cereal crop yields of 15–89% have been achieved as a result of deep ripping (subsoiling) sandplain soils in WA [15–17] which is consistent with the magnitude of cereal crop responses to deep ripping measured on similar soil types in overseas studies [18,19]. Similarly average yield increases, in the order of 16–50%, have been found where water repellence has been alleviated through the addition of subsoil clays [17], furrow sowing [20], sowing back onto or near previous seeding lines [10,21], dilution and burial of water repellent topsoils through strategic inversion tillage [22–24] and the addition of soil wetting agents [20,25].

However, where there are multiple limiting factors of a similar yield-limiting magnitude the gains from ameliorating one factor will be small relative to the gains where multiple constraints are ameliorated together [26]. For instance, ameliorating water repellence alone through the addition of clay, while increasing grain yields by more than 30% still resulted in crops yielding substantially below their calculated water-limited yield potential [27]. Strategic inversion and deep mixing of water repellent topsoils through mouldboard or one-way disc ploughing and rotary spading have been shown to be highly effective in reducing water repellence in Western and South Australian soils [23,28]. This process, which is termed strategic tillage [24,29–31], is a single or occasional use of tillage within what is otherwise a conservation agriculture farming system encompassing minimum tillage seeding and stubble retention. Strategic tillage potentially removes multiple soil, pest and weed limitations in a single operation with improvements in crop yields persisting for several years [23,30,32]. Strategic tillage can take the form of shallow or deep cultivation [23,32,33] depending on the soil type and nature of the constraints being addressed. Strategic deep soil mixing or inversion will de-compact the soil and incorporate soil amendments such as lime to working depth of the implement [24]. The aim of this paper is to assess combinations of treatments including claying, liming and strategic deep tillage that alleviate multiple and single soil constraints including water repellence, subsoil compaction, acidity, and nutrient deficiencies on a common sandplain soil on the south coast of WA and to evaluate their effects on crop productivity over 5 successive cropping seasons.

## 2. Materials and Methods

### 2.1. Experimental Sites

A field experiment was established at the Department of Primary Industries and Regional Development Esperance Downs Research Station (S33.608°, E121.785°) in June 2012 on a pale, deep non-gravelly sand overlying a sodic kaolin-dominant horizon at a depth of 0.8 m. The climate

is defined by temperate wet winters (June–August) and dry hot summers (December–February). Mean monthly temperatures range from 10–22 °C with an average annual rainfall of 492 mm of which 73% falls from April through to October. The site had been in volunteer pasture for the previous 15 years and had previously been deemed unsuitable for agronomic research due to the severity of the soil constraints. The soil is classified as an Albic Arenosol [13], and yellow mesonatric Sodosol [34] within the World Reference Base and Australian classification system, respectively. Soil chemical and physical properties (Table 1) show that the soil limitations to crop production include root limiting soil strength (cone index (CI) > 1500 kPa), severe water repellence (molarity of ethanol droplet test (MED) > 3) and low levels of extractable potassium ($K_{Col}$ < 40 mg $kg^{-1}$), organic carbon (OC) and cation exchange capacity (CEC). Soil pH was marginal for acidity limitation ($pH_{CaCl2} \geq 4.8$) through the 0.1–0.4 m depth (Table 1).

**Table 1.** Initial soil properties including soil strength (cone index, CI), water repellence using the molarity of ethanol droplet test (MED), electrical conductivity (EC), organic carbon (OC), $pH_{CaCl2}$, exchangeable Al, extractable P, K and S and cation exchange capacity (CEC) for the deep sand profile and the clay-rich subsoil material applied to the clay treatments at the experimental site.

| Depth | CI | MED | OC | $pH_{CaCl2}$ | P | K | S | Al | CEC |
|---|---|---|---|---|---|---|---|---|---|
| m | kPa | | % | | | mg $kg^{-1}$ | | cmol (+) $kg^{-1}$ | |
| 0–0.1 | 1530 | 3.1 | 1.3 | 5.2 | 12 | 38 | 16 | 0.05 | 3.0 |
| 0.1–0.2 | 2690 | 0 | 0.6 | 4.9 | 8 | 22 | 9 | 0.10 | 1.3 |
| 0.2–0.3 | 3050 | 0 | 0.3 | 4.8 | 13 | 15 | 4 | 0.11 | 0.9 |
| 0.3–0.4 | 2540 | 0 | 0.1 | 4.9 | 17 | 16 | 4 | 0.17 | 0.7 |
| 0.4–0.5 | 2030 | | 0.1 | 5.0 | 26 | 18 | 4 | 0.20 | 0.8 |
| 0.5–0.7 | | | 0.1 | 5.2 | 11 | 26 | 4 | 0.29 | 0.9 |
| Applied clay rich subsoil | 0 | | 0.1 | 8.0 | 2 | 1090 | 23 | <0.001 | 14.7 |

Eighteen treatments were imposed at the experimental site and replicated four times to give 72 plots. The experiment had nine main treatments and two lime sub treatments in a split plot design with four randomised blocks (Table 2).

**Table 2.** Treatment descriptions including implements, depths and application rates.

| No | Main Treatment | Treatment Code | Lime Rate |
|---|---|---|---|
| 1. | Control—knife point | Control | ±Lime 2 t $ha^{-1}$ |
| 2. | Wetting Agent (8 L $ha^{-1}$) | Wetter | ±Lime 2 t $ha^{-1}$ |
| 3. | Seeding system—Disc | Disc | ±Lime 2 t $ha^{-1}$ |
| 4. | Seeding system—Winged point | WingedP | ±Lime 2 t $ha^{-1}$ |
| 5. | Seeding system—Paired row | PairedR | ±Lime 2 t $ha^{-1}$ |
| 6. | Spader to 0.35 m | Sp35 | ±Lime 2 t $ha^{-1}$ |
| 7. | Mouldboard plough to 0.35 m | Mbd35 | ±Lime 2 t $ha^{-1}$ |
| 8. | Spader 0.2 m + Clay 170 t $ha^{-1}$ | Clay_Sp20 | ±Lime 2 t $ha^{-1}$ |
| 9. | Spader 0.35 m + Clay 170 t $ha^{-1}$ | Clay_Sp35 | ± Lime 2 t $ha^{-1}$ |

The plots were 4.5 m wide and 20 m long. A sub-treatment of lime sand with a neutralising value of 70% was surface spread at a rate of 2 t $ha^{-1}$. The lime was spread prior to the tillage and clay treatments as an 18 m wide strip splitting all 36 main treatment plots within the experiment. The lime treatment was not randomised within the blocks or treatments. Subsoil clay (Table 1) excavated from a nearby farm was applied at a rate of 170 t $ha^{-1}$ in two of the treatments. The rate applied was consistent with increasing the clay content of the 0.1 m topsoil layer to 5% which is required to ameliorate water repellence in soils with less than 2% organic carbon [17,35,36]. The applied subsoil comprised 70% clay (<2 μm) fraction and was predominately kaolinite with traces of illite and glauconite. Evenly spaced bucket loads of subsoil were deposited and then spread uniformly across the plot with the skid loader

bucket used as a grader blade. The weight of subsoil applied was measured by the load (approximately 300 kg load$^{-1}$) and checked periodically using a weigh bridge. The spading and mouldboard ploughing treatments were applied with a Farmax Rapide® Rotary Spader (Denekamp, The Netherlands) and a John Deere 995® (Moline, IL, USA) on-land reversible mouldboard square plough, respectively. The Spader was used to incorporate the clay treatments to depths of 0.2 or 0.35 m. The working depth for the mouldboard plough was 0.35 m. The lime, clay and tillage treatments were applied once only for the duration of the experiment whereas the wetting agent and seeding boot treatments were applied annually at seeding.

The experiment was sown with a 1.5 m wide cone plot seeder requiring three passes per plot. Four types of seeding points were fitted to the cone seeder tynes at seeding. The seeding points ranged from least to most soil disturbance in the order: Disc seeder (Great Plains, Salina, KA, USA, treatment 5) < Knife points (Primary Sales, Perth, Australia, treatments 1, 2, 6–9) < Winged point super seeder (Primary Sales, Perth, Australia, treatment 4) < Paired row (Stiletto®, Geraldton, Australia, treatment 3). Precision Wetter® (Chemsol Australia Pty. Ltd., Perth, Western Australia), a non-ionic soil wetting agent was applied by boomspray at seeding at a rate of 8 litres ha$^{-1}$ in 100 L of water ha$^{-1}$ as a blanket application. Nitrogen, P and K fertilizers were drilled at seeding while S was broadcast as gypsum prior to seeding. Nitrogen was also applied at tillering as liquid urea-ammonium nitrate (CSBP Flexi-N®, Perth, Australia) or as Urea. Potassium, a limiting nutrient in these soils, was applied only in the years when canola was grown. The rationale for this was to test the effects of the clay-rich subsoil on K nutrition in cereals. Pesticides, including herbicides, fungicides and insecticides were applied uniformly across the experiment to optimise crop yields in each season. The canola and cereal crops were mechanically harvested with a small plot harvester in mid-November and late-November through to early December, respectively. Details of the cropping rotation, agronomic management, annual rainfall, rainfall deciles and calculated yield potential [37] are given in Table 3. Rainfall deciles rank annual rainfall from the lowest 10% (Decile 1) to the highest 10% (Decile 10) of prior annual (1952–2016) rainfall records. Rainfall limited yield potential were calculated for wheat and barley using rainfall use efficiency terms of 20 kg mm$^{-1}$ ha$^{-1}$ [37] and 15 kg mm$^{-1}$ ha$^{-1}$ for canola [38].

**Table 3.** Crop species, varieties, seeding rates, fertilizer applications, seeding and harvest dates, annual rainfall total, rainfall decile and calculated potential yield for the experimental site between 2012 and 2016.

| | Year | | | | |
|---|---|---|---|---|---|
| **Variable** | **2012** | **2013** | **2014** | **2015** | **2016** |
| Crop | Wheat | Canola | Barley | Canola | Wheat |
| Variety | Mace | Henty | Bass | Wahoo | Mace |
| Seeding date | 5 July | 13 May | 27 May | 5 May | 20 May |
| Seeding rate kg ha$^{-1}$ | 90 | 5 | 103 | 5 | 90 |
| Fertilizer N:P:K:S kg ha$^{-1}$ | 39:14:0:9 | 63:10:29:10 | 32:9:0:13 | 116:9:60:20 | 56:14:0:43 |
| Ann Rainfall mm | 511 | 626 | 410 | 470 | 539 |
| Rainfall Decile | 6 | 10 | 2 | 4 | 7 |
| Rainfall limited yield potential (t ha$^{-1}$) | 3.8 | 3.3 | 3.2 | 2.7 | 4.0 |

*2.2. Soil Chemical and Physical Properties*

Topsoil water repellence was measured using the Molarity of Ethanol Droplet (MED) test [39] on soils collected to a depth of 0.1 m that had been oven dried (<70 °C) for 48 h and allowed to cool prior to measuring. The rating categories for the MED test are; low 0–1.0, moderate 1.2–2.0, severe 2.4–3.0 and very severe >3.2 [39]. Only the control, spaded (±clay) and mouldboard ploughed treatments were measured in 2014 whereas all treatments were measured for water repellence in 2015 and 2016. Soil strength was measured annually in mid-winter when the soils were at field capacity using a RIMIK® (CP40-II, Toowoomba, Australia) digital recording cone (130 mm$^2$ ASAE standard S313.3) penetrometer. Triplicate penetration resistance readings were measured to 0.6 m depth in

three locations within each plot, with strength recorded every 0.01 m. Soil water measurements were collected to a depth of 0.12 m with a Campbell HydroSense II® (Logan, UT, USA) handheld soil moisture sensor in five locations within each plot when crops were emerging.

Soil chemical properties were measured on samples that were collected in June 2017 from the control, mouldboard ploughed and spaded (±clay) treatments with and without lime. Samples were collected at 0.1 m depth increments to 0.4 m at ten sites within each plot using a 0.05 m OD auger. Samples for each depth within each plot were bulked together and split in half. The methods for; electrical conductivity (EC), soil pH ($pH_{CaCl2}$) and Aluminium ($Al_{CaCl2}$) measured in 0.01 M $CaCl_2$ with a soil to solution ratio of 1:5, Walkley-Black organic carbon (C), exchangeable cations (Al, Ca, Mg, K, Na) extracted in 0.1 M $NH_4Cl$, Colwell bicarbonate phosphorus ($P_{Col}$) and K ($K_{Col}$) and 1 M KCl-extractable nitrate and ammonium are described by [40]. Cation exchange capacity (CEC) was calculated as the sum of exchangeable cations. Sulfur ($S_{KCl40}$) was extracted in 0.25 M KCl using the method of [41]. Copper (Cu), zinc (Zn) and manganese (Mn) was extracted using DTPA solution [40]. Particle size analysis was done using the pipette method of [42].

*2.3. Plant Measurements*

Plant numbers were counted post seeding along adjacent 1 metre rows in five locations within each plot. Normalised difference vegetation index (NDVI) measurements were recorded prior to ear emergence (cereals) or flowering (canola) using a Trimble Greenseeker® (Sunnyvale, CA, USA). Biomass samples were collected annually in each plot at growth stages near anthesis for the cereals or prior to pod filling for canola (early September). Plant samples were harvested from three 0.5 m$^2$ sites within each plot and bulked together. Dry weights and tiller numbers were measured on plant samples that had been dried at 70 °C for 3 to 4 days in a fan forced oven. Grain yield (t ha$^{-1}$) was measured in November or December using plot harvesters.

Weed numbers were counted within 3 by 0.5 m$^2$ quadrats per plot in late August 2012 and late June 2013. The weed species present were not counted separately but were dominated by rye grass (*Lolium rigidum G.*), brome grass (*Bromus diandrus R.*) and Erodium (*Erodium cicutarium L.*) with minor numbers of wild radish (*Raphanus raphanistrum L.*) and volunteer clovers (*Trifolium subterraneum L.*).

Youngest fully expanded leaf (2012) and whole top (2013–2016) biomass subsamples were analysed by ICP for macro and micro nutrients [43] at the CSBP laboratory (Perth, Australia). Plant tissue analysis was performed only on the control, mouldboard ploughed and spaded (±clay) treatments with and without lime. Grain yield was measured by mechanically harvesting two 20 m by 1.8 m wide strips within each plot. Grain samples were collected from each plot and analysed for seed size and weight. An Infratech™ (Hillerød, Germany) grain analyser was used to measure colour, moisture, oil content and protein. Grain yields reported were not adjusted for moisture.

A partial nutrient balance for N, P, K and S were calculated over the five years of the experiment. The partial nutrient balance consisted of inputs (fertilizer) minus outputs (N, P, K and S within the grain removed) [44]. The inputs were the same for each treatment. The outputs were calculated by multiplying grain yields by published nutrient levels [45] of wheat, canola and barley.

*2.4. Economics*

A discounted cash flow approach was used to assess the profitability of each treatment. Gross margins (Grain yield t ha$^{-1}$ × grain price AUD t$^{-1}$ − variable costs AUD ha$^{-1}$) were recorded for each year. The variable costs for seed, fertilizer and pesticides amounted to AUD 220 ha$^{-1}$. The cost of the Wetter treatment (AUD 40 ha$^{-1}$) was added to the variable costs each year. Costs for the Paired row (AUD 10 ha$^{-1}$), mouldboard ploughing (AUD 110 ha$^{-1}$), spading to 0.35 m (AUD 150 ha$^{-1}$), clay addition with spading to 0.2 m (Clay_Sp20, AUD 775 ha$^{-1}$) and clay addition with spading to 0.35 m (Clay_Sp35, AUD 850/ha) were added to the variable costs in the initial year only. The strategic tillage costs are based on those reported by [46,47]. A 5% discount rate was used to determine the

Net Present Value (NPV) for each treatment. Return on Investment (ROI) was calculated as the (NPV Profit—NPV treatment cost/NPV treatment cost).

*2.5. Statistics*

Statistical differences among treatments were assessed using the ANOVA function within Genstat [48]. The nine main treatments were analysed individually whereas the lime sub-treatment was analysed as a non-randomised split plot within the main treatments. Data for the limed treatments are presented only where there was a significant interaction between the tillage main treatments and the limed sub-treatments. Penetrometer and soil chemical measurements were statistically compared between the treatments at individual depths. Linear correlations and variation accounted for ($r^2$) between variables were made using the trend function in Excel 2010 (Microsoft Corp, Redmond, WA, USA).

## 3. Results

*3.1. Water Repellence*

Topsoil water repellence at the site was rated as severe to very severe with MED values for the control treatment ranging from 2.8–3.6 (Figure 1) in all years. Spading, mouldboard ploughing and the clayed treatments reduced topsoil water repellence ($p \leq 0.05$) when compared to the control. Spading alone was rated as having a moderate level of repellence which was higher than the mouldboard ploughed and clayed treatments which were rated as having low levels of repellence. The wetting agent and the seeding point treatments did not change topsoil water repellence when compared to the control at the times the measurements were made. Lime application had no effect on water repellence.

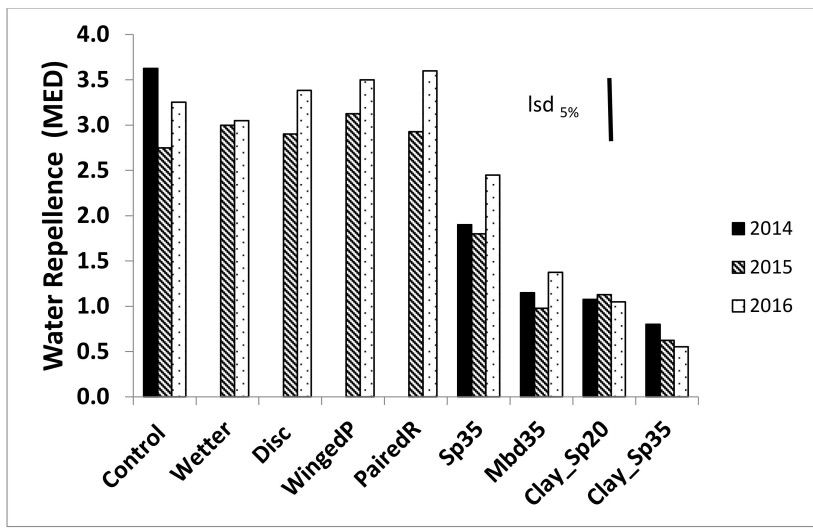

**Figure 1.** Effect of the tillage main treatments on water repellence as measured by the molarity of ethanol droplet (MED) test on soils collected to a depth of 0–0.10 m. Measurements made during 2014, 2015 and 2016 prior to seeding and wetter application. The LSD ($p \leq 0.05$) shown is the maximum value calculated over the three years. Treatment abbreviations are listed in Table 2.

*3.2. Soil Chemistry*

Soil $pH_{CaCl2}$ ranged from 5.9 in the 0–0.1 m layer to 4.8 within the subsoil (0.1–0.35 m) (Figure 2A). When averaged across the main treatments, the addition of lime had a negligible effect on soil $pH_{CaCl2}$ to a depth of 0.2 m but did increase pH by 0.1 units within the 0.2–0.4 m layer (Figure 2A,B).

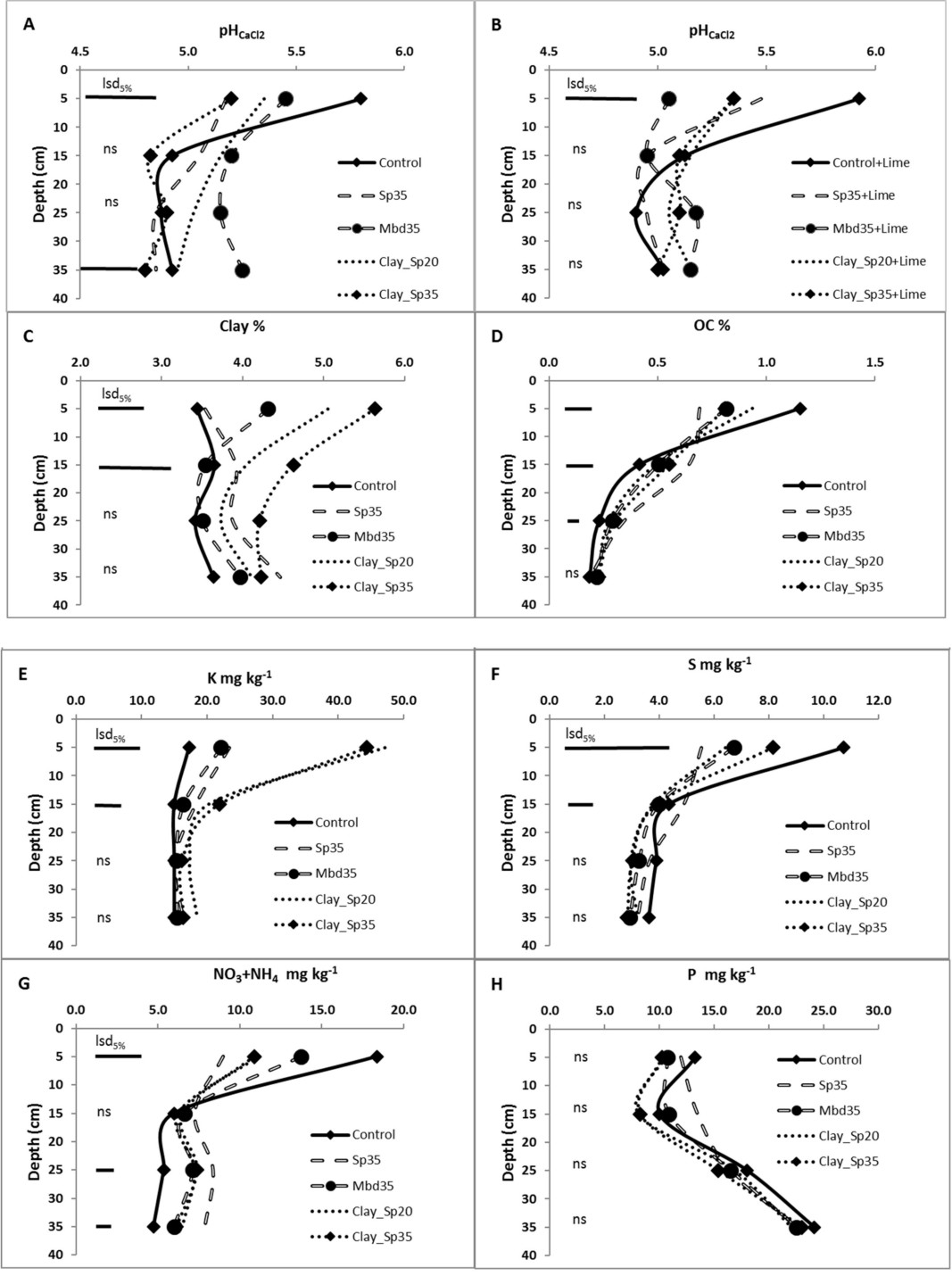

**Figure 2.** Effect of tillage and clay treatments on soil pH$_{CaCl2}$ without lime (**A**), soil pH$_{CaCl2}$ with lime added (**B**) clay % (**C**), organic carbon (OC%) (**D**), sum of nitrate and ammonium (mg kg$^{-1}$) (**E**) extractable S (mg kg$^{-1}$) (**F**) Colwell-extractable K (K$_{Col}$)(mg kg$^{-1}$) (**G**) and Colwell-extractable P (P$_{Col}$) (mg kg$^{-1}$) (**H**) at four depths 0–0.1, 0.1–0.2, 0.2–0.3 and 0.3–0.4 m. Data collected in June 2017. Treatment abbreviations are listed in Table 2. Horizontal bars represent the LSD ($p \leq 0.05$) for the corresponding depth; ns- not significant ($p \leq 0.05$).

The Control treatment had higher pH$_{CaCl2}$, OC%, S and mineral nitrogen (NO$_3^-$ + NH$_4^+$) in the 0–0.1 m layer than the Mbd35 and Sp35 treatments (Figure 2A,B,D,F,G). At lower depths in the soil profile, the Control had lower pH$_{CaCl2}$, OC% and NO$_3$+ NH$_4^+$ than the Mbd35 treatment and similar

values to the Sp35 (Figure 2D,G). Clay content and K were increased to more than 5.2% and 44 mg kg$^{-1}$ to a depth of 0.1 m respectively within the Clay_Sp20 and Clay_Sp35 treatments compared to less than 3.5% and 20 mg K kg$^{-1}$ for the Control and Mbd35 treatments (Figure 2C,E). Below 0.1 m depth there were no significant differences in clay content between the treatments (Figure 2C). At depths greater than 0.2 m there were no differences in K among treatments. The Colwell-extractable P was not affected by the tillage and clay treatments at any depth measured (Figure 2H).

### 3.3. Soil Strength

Soil penetration resistance increased with depth achieving maximum strength at a depth of 0.3 m (Figure 3). In each year, soil strength in the Control treatment exceeded 2500 kPa at 0.19 m depth and 3000 kPa at 0.22 m depth. Soil strength was significantly reduced as a result of the Spading (±Clay) and Mbd35 ploughed treatments when compared to the Control (Figure 3A). The reductions in soil strength occurred within the 0.05–0.32 m zone in the initial year and in later years between the depths of 0.1 and 0.32 m (Figure 3B–D). In 2013 and 2014, soil strength was reduced in the order; Control > Clay_Sp20 > Mbd35 > Clay_Sp35 = Sp35 (Figure 3A,B). However, by 2016 the order in which the treatments reduced soil strength had changed to Control = Clay_Sp20 > Mbd35 = Sp35 = Clay_Sp35 (Figure 3D). There were no differences between the limed and unlimed treatments and no consistent differences among the Control, Wetter, WingedP and PairedR sowing treatments.

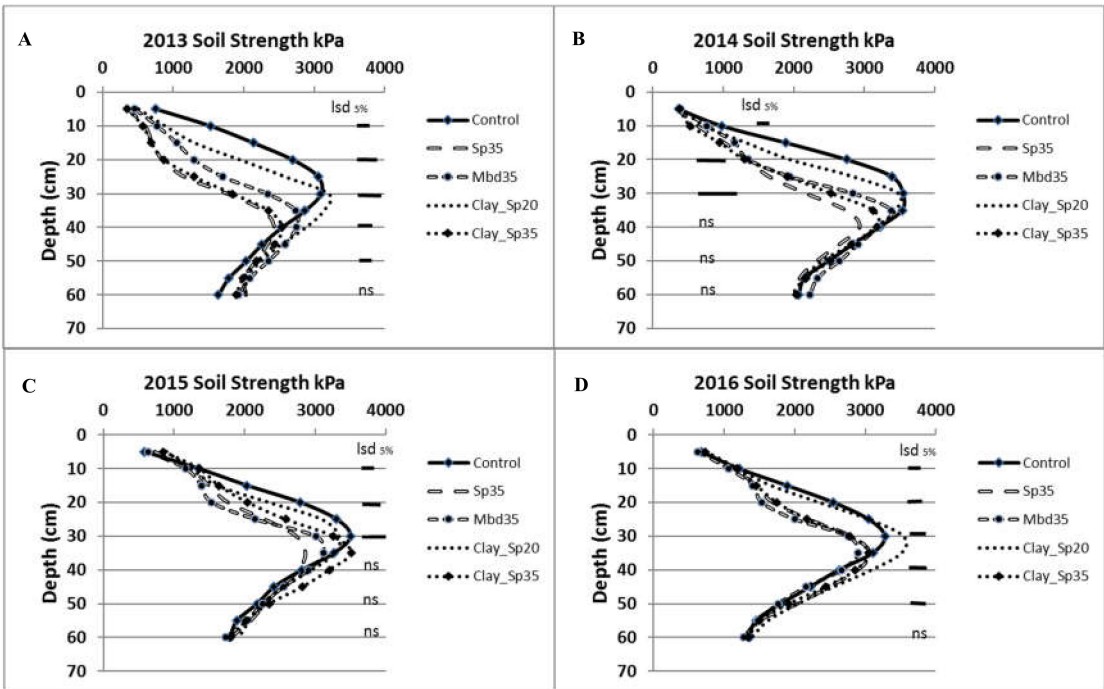

**Figure 3.** Soil strength as affected by the control, spaded (±clay) (Sp35) and mouldboard ploughed (Mbd) treatments. Measurements recorded during the winter months of 2013 (**A**), 2014 (**B**), 2015 (**C**) and 2016 (**D**). Treatment abbreviations are listed in Table 2. Horizontal bars represent the LSD ($p \le 0.05$) for the corresponding depth; ns- not significant ($p = 0.05$).

The differences in soil strength between the control and the spaded and mouldboard ploughed treatments diminished with time (Figure 4). Initially the Control had soil strength values up to 1800 kPa higher than the Sp35 treatment in 2013 at a depth of 0.2 m. However, by 2016 the corresponding difference was 860 kPa (Figure 4A). The reduction in soil strength for the Mbd35 treatment when compared to the Control (Figure 4B), was maintained over time more so than the Sp35 treatment (Figure 4A). Both the Mbd35 and Sp35 had minor ($p > 0.2$) but consistent increases in soil strength compared to the Control immediately below the plough layer.

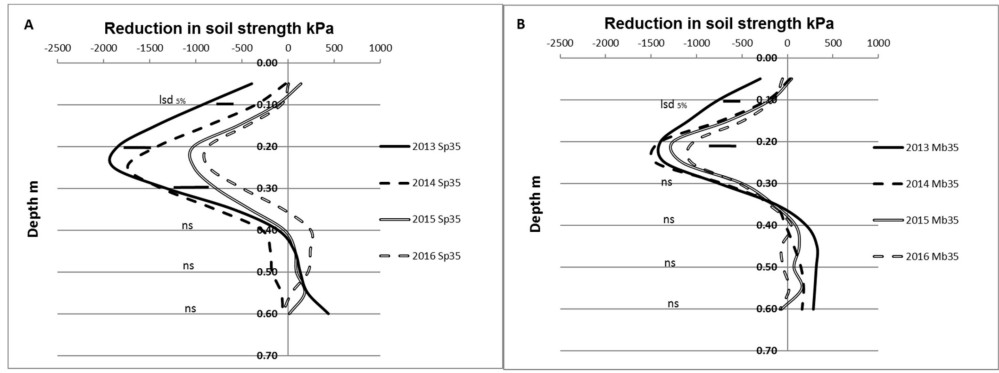

**Figure 4.** Reduction in soil strength between the Control and the treatment spaded to 35 cm depth (Sp35) (**A**) and between the Control and the treatment mouldboard ploughed to 35 cm depth (Mbd35) (**B**) to 60 cm depth for each year (2013–2016). Positive values occur where soil strength for the Sp35 and Mbd35 treatments exceed that of the Control. Horizontal bars represent the LSD ($p \leq 0.05$) for the corresponding depth; ns- not significant ($p \leq 0.05$).

### 3.4. Plant Emergence

Plant numbers exceeded 115 and 35 plants m$^{-2}$, the minimum thresholds to reach yield potential for cereals and canola, respectively, in each year irrespective of treatment. The PairedR, and Wetter treatments had inconsistent effects on crop emergence with increased and reduced plant numbers in 2014 and 2016, respectively, when compared to the control (Table 4). The WingedP treatment increased plant numbers in 2012, 2014, 2015 but reduced them in 2016 when compared to the control. The Mbd35 treatments reduced canola plant numbers in 2013 while the Sp35 treatment increased plant numbers at establishment in 2014 and 2015 when compared to the control.

**Table 4.** Crop establishment (plants m$^{-2}$) for wheat (2012, 2016), canola (2013, 2015) and barley (2014). Treatment abbreviations are listed in Table 2.

| Treatment | 2012 | 2013 | 2014 | 2015 | 2016 |
|---|---|---|---|---|---|
| **Crop** | **Wheat** | **Canola** | **Barley** | **Canola** | **Wheat** |
| Control | 158 ab | 79 b | 116 a | 38 a | 151 c |
| Disc [+] | 142 ab | NA | NA | NA | NA |
| WingedP | 182 c | 74 b | 135 b | 41 bc | 130 ab |
| PairedR | 153 ab | 75 b | 145 b | 37 ab | 119 a |
| Wetter | 162 bc | 74 b | 121 a | 31 a | 136 ab |
| Sp35 | 141 ab | 76 b | 139 b | 43 bc | 148 bc |
| Mbd35 | 144 ab | 58 a | 116 a | 40 b | 140 b |
| Clay_Sp20 | 154 ab | 76 b | 134 b | 54 d | 160 c |
| Clay_Sp35 | 139 a | 73 b | 145 b | 51 cd | 160 c |

[+] Mechanical problems with the disc seeder prevented even establishment in the years after 2012. NA measurement not available. Means in each column followed by the same letter were not significantly ($p < 0.05$) different.

The clay treatments (Clay_Sp20 and Clay_Sp35) increased barley and canola emergence in 2014 and 2015, respectively, when compared to the control (Table 3). Volumetric soil water contents in the clayed treatments (5.5 to 6.5%) within the 0–0.12 m layer were significantly higher ($p \leq 0.05$) than the Control (4.4%) in 2015 (data not presented). Seven of the eight significant treatment increases in plant emergence when compared to the Control occurred in years where rainfall was less than Decile 5. There were no significant interactions between the tillage and lime treatments with the means for the lime treated and untreated plots varying by less than four plants per square metre in each year. Water repellence, as measured by the MED test, accounted for 42% of the variation in emergence of canola in 2015. In all other years, MED accounted for between 1 and 22% of the variation in plant

emergence. Similarly, plant density at emergence accounted for less than 26% (range 1–26%) of the variation in grain yield in any year.

Weed populations ranged from 5–26 plants m$^{-2}$ in 2012. Weed populations were reduced ($p < 0.05$) in the order; Clay_Sp20 > Control = Clay_Sp35 > Disc = Mbd35. In June 2013 weed populations ranged from 5–12 plants m$^{-2}$ across all treatments with the Mbd35 and Sp35 being the only treatments to have lower ($p < 0.05$) weed numbers (5 plants m$^{-2}$) than the control (10 plants m$^{-2}$). Lime had no effect on weed numbers.

### 3.5. Crop Development and Grain Yields

Wheat yields in 2012 were increased as a result of the mouldboard ploughing and spading (±clay) treatments when compared to the control (Table 5). The remaining treatments did not differ significantly to the control. The deep incorporation of clay (Clay_Sp35) gave no yield benefit over the deep incorporation without clay (Sp35). A significant interaction between the main treatments and the lime sub treatment occurred resulting in the limed sub treatments consistently yielding 0.2–0.6 t ha$^{-1}$ more than the unlimed treatments. Overall, the grain yields for 2012 were lower than the calculated potential (3.9 t ha$^{-1}$) due to late sowing in early July and the presence of Take All root disease (*G. graminis*) affecting 20–40 % of the grain heads sampled.

**Table 5.** Grain yields (t ha$^{-1}$) for wheat (2012, 2016), canola (2013, 2015) and barley (2014) as affected by the main treatments. Treatment abbreviations are given in Table 2.

| Treatment | 2012 | 2013 | 2014 | 2015 | 2016 | Total |
|---|---|---|---|---|---|---|
| Crop | Wheat | Canola | Barley | Canola | Wheat | |
| Control | 1.01 a | 2.08 a | 2.04 a | 1.81 abc | 4.32 ab | 11.2 a |
| Disc | 1.18 ab | NA | NA | NA | NA | NA |
| PairedR | 0.96 a | 2.33 ab | 2.12 a | 1.94 ab | 4.28 ab | 11.6 a |
| Wetter | 1.08 a | 2.08 a | 1.84 a | 1.42 ab | 3.91 a | 10.6 a |
| WingedP | 1.01 a | 2.15 ab | 1.93 a | 1.31 a | 4.15 a | 10.3 a |
| Sp35 | 1.71 d | 2.22 ab | 3.08 c | 1.94 abc | 4.72 bc | 13.7 b |
| Mbd35 | 1.74 d | 2.33 abc | 2.92 c | 2.30 c | 4.55 bc | 13.8 b |
| Clay_Sp20 | 1.46 cd | 2.57 c | 2.56 b | 2.15 c | 4.64 bc | 13.4 b |
| Clay_Sp35 | 1.43 bc | 2.38 bc | 2.97 c | 1.99 bc | 4.91 c | 13.7 b |

NA measurement not available. Means in each column followed by the same letter were not significantly ($p < 0.05$) different.

Early season NDVI measurements in 2013 showed canola vigour declined in the order of: Control (0.26) > remaining treatments (0.23–0.2) > Sp35 (0.17) = Mbd35 (0.15). Despite this, canola yields in 2013 were increased by both the clayed and spaded treatments (Clay_Sp20 and Clay_Sp35) when compared to the Control (Table 5). All other treatments had grain yields which were similar to the control. The highest treatment yield (Clay_Sp20) attained 78% of the calculated rainfall-limited yield potential. Annual rainfall for the year was exceptionally high (Decile 10) with approximately a third of this occurring prior to seeding.

Barley grain yield in 2014 was increased by the spading, mouldboard ploughing and the clayed treatments when compared to the control (Table 5). The increase in grain yield associated with these treatments ranged from 0.5 to 1.0 t ha$^{-1}$. All other treatments yielded similarly to the control. The deep spading with clay (Clay_Sp35) gave no additional yield benefit compared to the Sp20 treatment. The grain yield from the Sp35 treatment achieved 90% of the calculated rainfall limited potential yield whereas the Control achieved only 60%.

Canola NDVI values measured in late July 2015 were increased ($p < 0.01$) only by the Clay_Sp20 and Clay_Sp35 treatments when compared to the Control. However, by the end of the season no significant differences in canola yield were found between the control and all other treatments in 2015 (Table 5). The WingedP and Wetter treatments had lowest yields and were significantly lower than the

spaded, mouldboard ploughed and clayed treatments. Yields for the mouldboard ploughed treatment were almost 90% of the rainfall limited yield potential.

Wheat NDVI was increased by the Clay_Sp20 and Clay_Sp35 treatment in early August 2016 when compared to the control. Wheat yields in 2016 averaged 4.3 t ha$^{-1}$ across the experimental site. The Clay_Sp35 treatment increased ($p < 0.001$) wheat grain yields by 0.59 t ha$^{-1}$ when compared to the control in 2016. No other treatment differed significantly to the control (Table 5). There were no differences in grain yield between the spaded (±clay) and mouldboard ploughed treatments.

Overall, the spading with clay treatments (Clay_Sp20 and Clay_Sp35) had significantly higher grain yields than the control in four of the five years. The Sp35 and Mbd35 treatments resulted in yield increases over the control in 3 and 2 of the five years, respectively. The grain yields in all of the other treatments (Disc, PairedR, Wetter, WingedP) did not differ significantly to the control in any year. Total production over the five years was increased significantly by the spading (±clay) and mouldboard ploughing treatments by 2.1–2.6 t ha$^{-1}$ when compared to the control. When averaged across all years, the highest yielding treatments achieved 85% of the estimated potential yield compared to 65% for the Control. Grain yields for the Sp35 and Clay_Sp35 treatments were statistically similar in each year. There was no significant interaction between the seeding and tillage main treatments and lime in any year.

The percentage increase in crop yields associated with the spaded (Sp35, Clay_Sp20, Clay_Sp35) and mouldboard ploughed treatments appears to have diminished with time when compared to the Control. Initial yield increases exceeded 70% for the Sp35 and Mbd35 treatments in 2012 whereas by 2016 the yield increases had declined to no more than 9% higher than the Control. Small yield increases were also found between the Control and the Clay_Sp20, Clay_Sp35, Sp35 and Mbd35 treatments in 2013 which was an exceptionally wet (Decile 10) year.

*3.6. Crop Nutrition*

The Clay_Sp20 and Clay_Sp35 treatments increased shoot K concentrations in all years except 2012 when compared to the Control (Table 6). The control treatment had shoot K concentrations below or near the critical value (2% K) for cereal crops [49]. Soil K within the 0–0.1 m layer for the control treatment was less than 30 mg kg$^{-1}$ compared to greater than 45 mg kg$^{-1}$ where clay had been applied (Figure 2E). Based on regression analysis, shoot K explained on average 20% (range <0.01 to 0.34, n = 40) of the variation in cereal and canola yields. Increased shoot K concentrations were also found in three of the five crops (2014, 2015, 2016) for the Sp35 treatment when compared to the Control.

**Table 6.** Shoot nutrient concentrations as affected by the tillage and clay main treatments collected in each of the cropping years from 2012 to 2016. The plant part sampled was the flag leaf (2012) and whole shoots in all other years. Plant samples were collected in early September of each year. Treatment abbreviations are listed in Table 2.

| Year and Crop | Treatment | N % | P % | K % | S % | Ca % | B mg kg$^{-1}$ | Zn mg kg$^{-1}$ |
|---|---|---|---|---|---|---|---|---|
| 2012<br>Wheat | Control | 3.99 b | 0.34 c | 1.99 b | 0.27 b | 0.33 b | 6.3 a | 18.3 a |
| | Sp35 | 3.78 a | 0.30 a | 1.83 ab | 0.25 a | 0.35 b | 6.3 a | 16.8 a |
| | Mbd35 | 3.78 a | 0.30 a | 1.79 a | 0.25 a | 0.36 b | 6.8 a | 17.4 a |
| | Clay_Sp20 | 3.95 b | 0.32 b | 2.00 b | 0.27 b | 0.28 a | 18.9 b | 18.1 a |
| | Clay_Sp35 | 3.90 ab | 0.32 b | 1.93 ab | 0.26 ab | 0.27 a | 18.5 b | 17.0 a |
| | Prob | $p < 0.02$ | $p < 0.02$ | $p < 0.06$ | $p < 0.01$ | $p < 0.001$ | $p < 0.001$ | ns |
| | LSD ($p \leq 0.05$) | 0.14 | 0.02 | 0.16 | 0.02 | 0.04 | 1.17 | |
| 2013<br>Canola | Control | 2.75 a | 0.45 a | 2.74 a | 0.31 a | 1.52 a | 26.0 a | 29.7 b |
| | Sp35 | 3.00 a | 0.51 ab | 2.90 ab | 0.31 a | 2.09 b | 29.0 ab | 27.0 ab |
| | Mbd35 | 3.28 a | 0.54 b | 2.97 ab | 0.36 a | 1.94 b | 28.4 ab | 28.8 b |
| | Clay_Sp20 | 2.67 a | 0.46 a | 3.33 bc | 0.27 a | 1.76 ab | 32.5 b | 24.7 a |
| | Clay_Sp35 | 3.37 a | 0.50 ab | 3.53 c | 0.42 a | 1.75 ab | 32.2 b | 24.3 a |
| | Prob | ns | $p < 0.04$ | $p < 0.02$ | ns | $p < 0.03$ | $p < 0.04$ | $p < 0.02$ |
| | LSD ($p \leq 0.05$) | | 0.06 | 0.48 | | 0.34 | 4.38 | 3.88 |

**Table 6.** *Cont.*

| Year and Crop | Treatment | N % | P % | K % | S % | Ca % | B mg kg$^{-1}$ | Zn mg kg$^{-1}$ |
|---|---|---|---|---|---|---|---|---|
| 2014 Barley | Control | 1.77 a | 0.24 a | 1.74 a | 0.14 a | 0.27 a | 3.06 a | 24.8 a |
| | Sp35 | 2.03 a | 0.27 a | 2.01 b | 0.17 b | 0.47 b | 3.52 a | 25.3 a |
| | Mbd35 | 2.46 b | 0.29 a | 1.76 a | 0.19 b | 0.62 c | 3.48 a | 37.5 a |
| | Clay_Sp20 | 2.04 a | 0.25 a | 2.57 d | 0.17 b | 0.36 a | 5.07 b | 27.0 a |
| | Clay_Sp35 | 2.18 ab | 0.28 a | 2.35 c | 0.17 b | 0.35 a | 4.94 b | 27.8 a |
| | Prob | $p < 0.001$ | ns | $p < 0.001$ | $p < 0.001$ | $p < 0.001$ | $p < 0.001$ | ns |
| | LSD ($p \leq 0.05$) | 0.31 | | 0.20 | 0.02 | 0.09 | 0.98 | |
| 2015 Canola | Control | 2.89 a | 0.29 a | 2.08 a | 0.36 a | 1.11 bc | 20.5 a | 38.9 c |
| | Sp35 | 2.91 a | 0.29 a | 2.52 b | 0.30 a | 1.14 c | 23.3 b | 33.2 b |
| | Mbd35 | 3.37 a | 0.34 a | 2.66 b | 0.31 a | 1.19 c | 24.1 b | 37.9 c |
| | Clay_Sp20 | 2.70 a | 0.27 a | 3.18 c | 0.28 a | 0.91 a | 26.8 c | 26.5 a |
| | Clay_Sp35 | 3.05 a | 0.34 a | 3.18 c | 0.30 a | 0.97ab | 31.0 d | 30.8 b |
| | Prob | ns | ns | $p < 0.001$ | ns | $p < 0.002$ | $p < 0.001$ | $p < 0.001$ |
| | LSD ($p \leq 0.05$) | | | 0.34 | | 0.14 | 2.51 | 4.18 |
| 2016 Barley | Control | 2.15 a | 0.34 | 1.29 a | 0.17 a | 0.44 b | 2.92 a | 20.4 a |
| | Sp35 | 2.25 a | 0.35 | 1.76 b | 0.17 a | 0.37 b | 2.57 a | 21.7 a |
| | Mbd35 | 2.32 a | 0.38 | 1.83 b | 0.18 a | 0.41 b | 2.76 a | 22.1 a |
| | Clay_Sp20 | 2.27 a | 0.32 | 2.27 c | 0.16 a | 0.25 a | 3.21 a | 18.7 a |
| | Clay_Sp35 | 2.29 a | 0.33 | 2.37 c | 0.17 a | 0.29 a | 2.98 a | 17.4 a |
| | Prob | ns | ns | $p < 0.001$ | ns | $p < 0.001$ | ns | ns |
| | LSD ($p \leq 0.05$) | | | 0.26 | | 0.07 | | |

Means in each column followed by the same letter were not significantly ($p < 0.05$) different.

Both clay treatments had higher shoot B levels than the control in all but one year. The shoot B values for cereal crops in the control treatment were less than 5 mg kg$^{-1}$ in two years which is rated as marginal to deficient [49]. However, shoot B explained on average only 5% (range <0.01 to 0.15, $n = 40$) of the variation in cereal and canola yields.

Significant differences in shoot N, P and S concentrations were found among the treatments, however, they were not consistent. For instance, the Mbd35 treatment had reduced plant N, P, K and S shoot concentrations in 2012 and increased N, P, K, Ca, S and B in some but not all subsequent years when compared to the Control (Table 6). Similarly, there were no consistent differences in shoot nutrient concentrations associated with the Sp35 treatment. The Sp35 treatment had reduced N, P and S shoot concentrations in 2012; however, in the remaining years, there was no difference except for increases in S concentration in some, but not all, subsequent years when compared to the Control. The concentrations of N%, P% and S% within whole shoots were generally low but on average above the critical thresholds [49] for cereals (2%N, 0.2%P, 0.15%S) and canola (2.2%N, 0.28%P, 0.3%S).

The Clay_Sp20 and Clay_Sp35 treatments had lower Ca levels than the Control in three of the six years (Table 6). Despite this, the values of Ca were above the critical deficiency (0.25%) levels [49] in all treatments.

Zn concentrations in the shoots were reduced as a result of spading (+clay) in canola in 2013 and spading (±clay) in 2015 when compared to the Control. Shoot Zn concentrations were borderline adequate [49] in most years being greater than 0.3 mg kg$^{-1}$ and 0.15 mg kg$^{-1}$ for canola and cereals, respectively (Table 6). Soil Zn levels were reduced in the 0–0.1 m layer in the spaded, inversion ploughed and clayed treatments when compared to the control.

*3.7. Partial Nutrient Balance*

The cumulative partial nutrient balances for N, P, K and S were calculated as the difference between nutrient inputs (fertilizer) and outputs (nutrients removed in grain) then summed for each year (Figure 5). Given that fertilizer rates did not differ among the treatments it is not surprising that the highest yielding treatments had the lowest positive P, K and S balance. By the end of the experimental period, the differences in the nutrient balances between the highest (Mbd35) and lowest (Wetter) yielding treatment were 125 kg N, 17 kg P, 26 kg K and 21 kg S (Figure 5). Phosphorus was

the only nutrient to have inputs matching or exceeding outputs in most years (Figure 5B). Nitrogen balances were negative in most years (Figure 5A) while K and S balances were negative in three of the five years (Figure 5C,D). The cumulative nutrient balances for all nutrients decreased in the order: (Wetter, Control, WingedP, PairedR) ≥ Clay_Sp20 = (Clay_Sp35, Sp35, Mbd35).

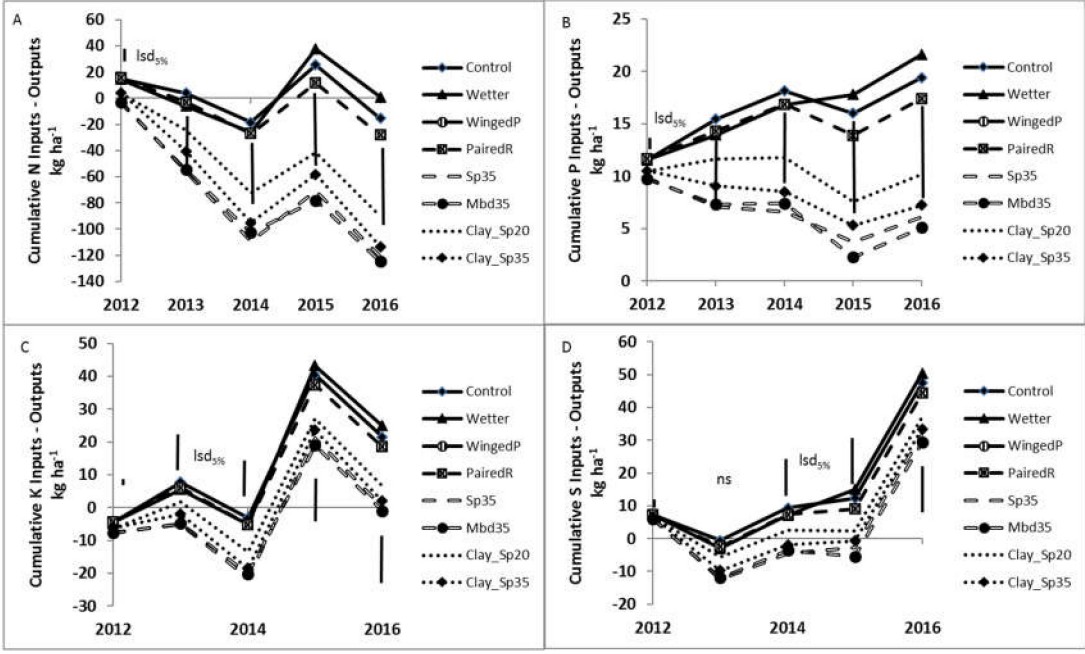

**Figure 5.** Treatment effects on cumulative nutrient balances (kg/ha) for Nitrogen (**A**), Phosphorus (**B**), Potassium (**C**) and Sulfur (**D**) for the years 2012 to 2016. Vertical bars represent the LSD ($p \leq 0.05$) for the corresponding year; ns is not significant ($p > 0.05$). Note K fertiliser was only added in 2012 and 2015. Treatment abbreviations are listed in Table 2.

### 3.8. Economics

Net present values (NPV) calculated over the five years of the experiment indicate that the most profitable treatments were the Mbd35, Sp35 and PairedR treatments which resulted in an increase in profit of AUD 653 ha$^{-1}$, AUD 457 ha$^{-1}$ and AUD 153 ha$^{-1}$, respectively, when compared to the Control. The Clay_Sp35, Clay_Sp20, Wetter and WingedP treatments were not as profitable as the Control (Table 7) in the short term. In terms of return on investment (ROI) the PairedR treatment gave the highest return (0.66). The equivalent ROI values for the Mbd35 and Sp35 treatments were 0.39 and 0.24, respectively.

**Table 7.** Effect of treatments on annual gross margins for the years 2012 to 2016.

| | | | | | AUD ha$^{-1}$ | | |
|---|---|---|---|---|---|---|---|
| | **2012** | **2013** | **2014** | **2015** | **2016** | **NPV5%** | **ΔProfit** |
| Control | 63 | 926 | 239 | 778 | 956 | 2495 | 0 |
| Wetter | 42 | 886 | 153 | 523 | 806 | 2037 | −458 |
| WingedP | 63 | 965 | 214 | 503 | 911 | 2246 | −249 |
| PairedR | 39 | 1064 | 247 | 849 | 938 | 2648 | 153 |
| Sp35 | 102 | 1003 | 478 | 849 | 1064 | 2952 | 457 |
| Mbd35 | 150 | 1064 | 442 | 1047 | 1019 | 3148 | 653 |
| Clay_Sp20 | −591 | 1196 | 359 | 965 | 1043 | 2442 | −53 |
| Clay_Sp35 | −674 | 1091 | 453 | 877 | 1116 | 2334 | −161 |

Net Present Value (NPV5%, AUD ha$^{-1}$) discounted at a rate of 5% and the increase in NPV over the Control (ΔProfit).

## 4. Discussions

The treatments selected were chosen primarily to alleviate water repellence which is prevalent on sandplain soils on the southern coast of WA. However, the expression of water repellence was mostly limited to 2014 and 2015 when there was lower than average annual rainfall (Decile 2 and 4), while the largest crop yield responses were to treatments that alleviated other limiting factors, compaction and K deficiency. On soils with multiple limitations, treatments which are able to alleviate all or part of each constraint will achieve the greatest productivity increase. However, when such treatments (e.g., clay addition), are expensive, they may not achieve a profitable response even within 5 years.

Water repellence develops on sandplain soils within the organically stained topsoil, but the degree of water repellence declines with depth to the extent that the subsoils are not water repellent. Diluting and burying the water repellent layer by mixing the top and subsoils using a spading machine or inverting the water repellent topsoils through mouldboard ploughing were effective in ameliorating water repellence as has been reported on a range of other sandy soils in WA [23,28]. The results presented here suggest that mouldboard ploughing, spading and the addition of clay rich subsoil had the largest effects on reducing water repellence when compared to the wetting agent and to seeding tynes with minimal to substantial soil disturbance. In this experiment, water repellence was not affected by the addition of lime. Elsewhere the addition of lime at rates of 5, 10 and 15 t ha$^{-1}$ to water repellent sandplain soils with a lower topsoil pH$_{CaCl2}$ of 4.9 has resulted in a significant reduction in MED values and a tenfold increase in the populations of wax degrading organisms [50].

Despite ploughing to 0.35 m, most of the changes in soil chemical properties occurred at depths less than 0.25 m. The distribution of clay in the profile was not affected by the different depths of spading with no changes in clay content below 0.15 m regardless of whether the soil was spaded to 0.20 or 0.35 m. This result is supported by [24] who found that the majority of the topsoil remained within the 0–0.15 m layer following spading to a depth of 0.25 m. The mouldboard ploughed treatment resulted in less mixing and greater burial of the topsoil given the lower organic carbon levels at the surface and higher organic carbon levels at 0.25 m than the spaded soils. This suggests that mouldboard ploughing is likely to be more effective in incorporating ameliorants to depth but is less likely to evenly distribute ameliorants when compared to spading. In soils with lower subsoil pH than the present site, the more effective incorporation of lime at 0–0.3 m depth should have significant benefits [51].

The addition and incorporation of clay-rich subsoils into sands has been shown to increase organic carbon, K and pH after 10 years in a field experiment on a similar sandplain soil to the present study [17] and reduced nitrate leaching in a pot experiment on sandy soil [52]. The increases in organic carbon are attributed to reduced microbial activity associated with clay addition [53] and increased biomass production [54]. However, no significant increases in organic carbon, mineral nitrogen or soil pH were found in this experiment after 5 years. Enhanced soil mixing associated with spading and the slow rates of carbon accumulation [55] in sands may explain this.

In each year of the experiment, cereal and canola plant establishment were adequate to achieve rainfall limited yield potential. Relationships between canola establishment and final yield have shown that 30–40 plants m$^{-2}$ (mean 32) is near the optimum to reach yield potential across a range of environments within WA [56]. The optimum plant density for wheat and barley can vary between 100 and 200 plants m$^{-2}$ [57]. However, in WA, where the growing season rainfall is greater than 161 mm the optimal wheat density was 124 plants m$^{-2}$ [58]. Overall crop establishment across all treatments was sufficient to achieve more than 85% of yield potential. The adequacy of plant establishment given the high MED values is attributed to two factors: favorable rainfall at seeding in most years and; seeds sown into the same rows for all crops in all years using ±0.02 m real time kinetic (RTK) guidance. On-row seeding has been shown to improve crop establishment in water repellent sands [10,59]. The poor relationships found between plant density at emergence and grain yield are further evidence of the adequacy of the established plant populations. Significant increases in plant densities were found in the majority of treatments only in the years (2014 and 2015) with below average rainfall. Canola emergence was found to be negatively affected by mouldboard ploughing in 2013. The reduction in canola

emergence within the first few years of mouldboard ploughing has also been observed in commercial canola crops. The reason for the poor emergence is currently being investigated with some evidence of increased herbicide activity that damages the crop in mouldboard ploughed soils associated with reduced topsoil organic matter [22,60,61]. In this experiment there was no consistent evidence to show that the knife point seeding boots were any less effective than the winged point or paired row seeding systems. Previous research has shown that knife point seeding tynes can exacerbate water repellence where the topsoil falls in behind the seeding tyne resulting in the seed being sown into a seam of water repellent soil [10,46].

The wetting agent and seeding tynes (WingedP, PairedR) did not significantly increase grain yields compared to the control in any year or in total. [25] summarized experimental data comparing grain responses to winged points, paired rows and wetting agents using data from experimental sites from the northern to southern wheatbelt. They found that the average grain yield increases attributed to winged points, paired rows and banded wetting agents were 6% (range −2 to 14%, *n* = 7), 19% (range 3 to 76%, *n* = 5) and 10% (range −32 to 32%, *n* = 19), respectively. The high degree of variability in grain yield response to banded soil wetting agents was further examined in a more recent review of changes in crop yield following implementation of strategies to manage soil water repellence [47]. This demonstrated that crop yield responses to banded wetting agents occurred when the crop was sown into dry repellent soils with negligible responses when the soil was wet. Cereal grain yield response to wetting agents was also related to soil type with an average 21% (*n* = 8, range 3–38%) yield increase with dry sowing on loamy gravel ([13], Ferralic Retisol) and duplex sandy gravel soils ([13], Abruptic Plinthosol), and 88% of comparisons showing statistically positive yield response. By comparison on deep sands ([13], Arenosols), the grain yield response to banded soil wetters on dry sown cereals averaged 12% (*n* = 6, range −13–32%) with only one-third of these comparisons having a statistically positive response [47]. In this experiment, the lack of response to soil wetting agents could largely be explained by the wet conditions at seeding in most seasons and a soil type that is typically less responsive to soil wetting agents.

Crop yields were increased as a result of the spaded, mouldboarded and clayed treatments by 19 to 23% when averaged over all years. Grain yield increases associated with spading and mouldboard ploughing have been consistently reported across a range of environments in Western and South Australia following many years of no-till crop establishment [23,62–64]. On average crop grain yields were found to increase by 51% (range −2 to 150%, *n* = 19) and 39% (range 0 to 100%, *n* = 11) for the mouldboarded and spaded treatments, respectively, in WA [65]. Results from a smaller number of South Australian studies [64] suggest even higher yields from spading with average grain yield increases of 111% (range 63 to 200%, *n* = 3). While spading was introduced to incorporate clay rich subsoils more effectively to remediate water repellence [28], the crop yield responses suggest that in the short term spading alone is as effective as spading clay-rich subsoil. Crop yield results from other experiments support this with no significant differences in grain yields found at three South Australian sites [64] and three sites in WA [65,66]. We therefore conclude that diluting the water repellent topsoil with the wettable subsoil is as effective as claying in the short term and in the absence of nutrient deficiencies. This is particularly evident in sands where there is a gradational increase in clay content with depth and where inversion tillage results in an increase in topsoil clay content.

The application of lime had no appreciable effect on crop emergence, grain yields or soil strength at this site. Apart from 2012, there were also no interactions between the lime and tillage treatments at this site. Soil pH exceeded $pH_{CaCl2}$ 4.8 throughout the soil profile, a value that is used as a threshold beyond which aluminium is unlikely to be present in its toxic form [67]. Hence there would be merit in follow-up studies on more acid soils with the lime treatment together with strategic deep tillage [51] and claying treatments.

The increases in yield occurred independently of crop emergence in all years. This suggests that the limitations imposed by water repellence on seedling emergence are secondary to other soil constraints at this site. Nutrient concentrations in plant tissue for N, P, K, S, and B were marginal in

most years (Table 6). Mouldboard ploughing and spading (−clay), although creating greater uniformity of nutrients within the soil profile, did not have an appreciable effect on nutrient levels in plant shoots. The clayed treatments increased K and B and reduced Ca in plant shoots. Elevated K levels in clayed soils have previously been proposed as a mechanism for increased yields in sandplain soils [17]. The increase in shoot K explained on average 19% (range 1–36%) of the variation in cereal and canola yields while effects from other nutrients were negligible. Elevated soil K levels can reduce Ca uptake [68] which would explain the reduced Ca levels found in the clayed treatments in three of the five years. Nutrient balances indicate that N, P, K and S decreased and were at times negative for the higher yielding treatments. Nitrogen balances were mainly negative for all treatments apart from the Control. The partial N balance, however, does not take into account N mineralised from organic carbon, or losses due to leaching. Despite this, the balances do show the importance of matching nutrition to higher production levels in modified soils. Based on the plant shoot and nutrient balance data, the contribution of nutrition to increased crop yields as a result of spading, mouldboard ploughing and claying is not conclusive given that N, P and S plant shoot concentrations did not differ between the Control and the highest yielding treatments in most years. Furthermore, the treatments with the highest shoot K concentrations (Clay_Sp20, Clay_Sp35) did not yield significantly higher than the Mbd35 and Sp35 treatments in four of the five years.

Although the experiment was managed to suppress weeds in all years, mouldboard ploughing had less than half the weed populations in 2012 and 2013 compared to the control whereas the results for spading (±clay) were more variable. Reductions in weed populations of up to 95% have been found as a result of mouldboard ploughing on sandy soils in Western Australia [69]. Spading is likely to be less effective at weed suppression given that the spading operation is not as effective at burying topsoil below 0.1m compared to mouldboard ploughing [24].

Soil strength exceeded 3000 kPa at a depth of 0.3 m in the Control treatment in all years measured. Values greater than 1500 kPa are likely to limit root growth while values greater than 3000 kPa are likely to stop root growth [70]. Few roots were observed below 40 cm in any treatment when sampled in 2017 [71]. Soil strength was reduced by the mouldboard ploughing and spaded treatments when compared to the Control in all years measured. Re-compaction of the mouldboard ploughed and spaded treatments resulted in diminishing differences in soil strength between these treatments and the Control with time. The absolute causes of the re-compaction are not known but are most likely due to machinery trafficking and natural soil wetting and drying cycles. Small plot spraying, seeding and harvesting equipment with axle loads less than 2.2 t followed the same tracks along the plots throughout the experiment. Such equipment, and the controlled traffic arrangement on plots would not be expected to re-compact soils beyond 0.25 m depth [72]. Natural forces, including cumulative rainfall [73], overburden pressure, and contraction of the soil matrix as water menisci evaporate during drying cycles [7], may also contribute to the re-compaction process in sands. That the largest yield increases in this experiment tended to have the lowest strength and that the differences in yield between the deep tilled treatments and the control have diminished over time, as has difference in soil strength, is unlikely to be a coincidence. In a review of deep tillage impacts on crop yield, [74] found that positive yield responses were associated with root-restricting layers, mostly compaction, in the soil profile and responses were greater in seasons with dry periods in the growing season. Hence, the present findings suggest that controlled traffic alone may not be sufficient to avoid re-compaction of soils such as those on the sandplains of southwest Australia, as suggested by [75] in their recently commentary.

The economic analysis has shown that the PairedR, Mbd35 and Sp35 treatments were the most profitable treatments over the 5-year period. In other studies, inversion ploughing with paired row seeding has given a AUD 33 ha$^{-1}$ increases in profitability in the first year after ploughing [62]. In a 3-year experiment, inversion ploughing increased the cumulative net financial return by AUD 777 ha$^{-1}$ and spading treatments by AUD 624–730 ha$^{-1}$ over the untreated control on a water repellent duplex sandy gravel ([13], Abruptic Plinthosol) [62]. The clayed treatments were not as profitable as the

control after five years. This is consistent with other studies that have shown claying to take up to seven years to break-even on sandplain soils [17].

## 5. Conclusions

Multiple constraints were ameliorated primarily by using strategic deep tillage and clay addition on sandplain soils in southern WA. Topsoil water repellence was reduced from being very severe-to-low as a result of the strategic deep tillage and applied clay treatments. There was no evidence at the end of five years to show that the water repellence was redeveloping following strategic deep tillage either with or without clay addition. Despite the reduction in water repellence, plant emergence was improved in only two of the five years. Both of these years (2014, 2015) had growing season rainfall at or below rainfall Decile 4 and affected barley and canola crops, respectively. Soil C, N and S were reduced in the 0.1 m layer while C and N were increased at lower depths as a result of the deep tillage treatments. Soil K was increased by the addition of clay and in most seasons the clayed treatment had higher shoot K concentrations than the control. Soil strength was reduced by the strategic deep tillage treatments and achieved penetration resistance values less than 2500 kPa to a depth of 0.3 m in the first year of the experiment. The finding that cumulative grain yield for the Sp35 treatment was identical to the Clay_Sp35 treatment suggests that K nutrition, inclusive of K applied in fertilizer at seeding, and topsoil water repellence were lesser soil constraints than compaction at this site. In successive years, the difference in penetration resistance between the deep tillage treatments and the control diminished to the extent that few roots were found below 0.4 m in any treatment [70]. This is mainly attributed to natural consolidation as opposed to machinery induced compaction. It is likely that the re-compaction process also resulted in reduced yield differences between the deep tillage treatments and the control over time. Given that high penetration resistance was found below 0.35 m, further increases in yield may occur with deeper soil loosening at this site. The strategic deep tillage and clay addition treatments increased cumulative crop yields by 2.1–2.6 t ha$^{-1}$ compared to the control over the five years of the experiment. Only the paired row, Sp35 and Mbd35 treatments were more profitable than the control. Indeed, the Mbd35 treatment increased the NPV by AUD 653 relative to the Control.

**Author Contributions:** D.J.M.H., S.L.D. and R.W.B. conceived the concepts of the manuscript; D.J.M.H. and T.J.E. performed the data collection; analysed the data and literature and wrote the manuscript. D.J.M.H., S.L.D. and R.W.B. obtained the funding. All authors have read and agreed to the published version of the manuscript.

**Funding:** We would like to thank Grains Research and Development Corporation and the Department of Primary industries and Regional Development for investing in this research through the projects DAW00244, DAW00242 and UMU00041.

**Acknowledgments:** The experiment was established with the help of local farmers, Nils Blumann, Greg Harris and Peter Luberda and the machinery dealership PH Kerr and Son. Invaluable technical support was provided by Chris Matthews and David Dodge.

**Conflicts of Interest:** The authors declare no conflict of interest.

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
