# Peer review of "Soil Management Systems to Overcome Multiple Constraints for Dryland Crops on Deep Sands in a Water Limited Environment on the South Coast of Western Australia"

_agronomy, doi:10.3390/agronomy10121881_

Round 1

Reviewer 1 Report

Huge and nice complex work with 5 years field studies.

I have some notes mainly in connection of typing problems and some questions:

Soil

Abstract

Line 22 and 23: No need any number in the abstract in connection with methods.

Introduction

Line 42: instead of km2 the correct is km2

Line 52: cation exchange correct unit is cmol(+) kg-1

Line 70: see the error with citation

Line 107 Table 1 Need to check the unit of Al and correct the unit of CEC, see above.

Line 116: How many plots per treatments did you have?

Line 191: citation is missing

Line 191: What do you mean on tissue analyses?

Line 251: Figure 2 Need to correct the unit on the figures as well (E, F, G, H) mg kg-1

Line296: Need to define int he title of table what does the small letter means?

Line 308: space problem 42%

Line 341: t ha-1 instead of t/ha

Line 393 and 394 enter problem + Line401 and 402 + Line 407 and 408 + Line 544 and 546 (connected to Table 6)

Line 410 Need to define this parameter correctly in the part of Materials and Methods chapter. Please add a reference for this parameter.

Line 416 Avoid starting the sentence with abbreviation. Phosphorous is better then P, and Line 417 with Nitrogen as well.

Line 424 Problem with kg/ha again, see all four figures + try to adjust the Y axe for each other, because different scales are badly influence the reader if it would like to compare them.

Line 426 P≤0.05, check all of the figures P<0.05 or that one?

Line 488 and 496 citation problems

Author Response

Dear Reviewer 1

Thank you for reviewing this paper. Your suggestions were very useful. Please see below in italics regarding my response to your suggested changes to the manuscript

Line 22 and 23: No need any number in the abstract in connection with methods.

Introduction . I have kept the numbers (depths and rates) in the abstract as I feel that they give the reader the detail required to understand the treatments applied.

Line 42: instead of km2 the correct is km2. Corrected in the text

Line 52: cation exchange correct unit is cmol(+) kg-1. Corrected in the text

Line 70: see the error with citation. Corrected in the text .

Line 107 Table 1 Need to check the unit of Al and correct the unit of CEC, see above. Corrected in the text and tables

Line 116: How many plots per treatments did you have? Plot numbers added in the text

Line 191: citation is missing. Citation added

Line 191: What do you mean on tissue analyses? “Plant” has been added to distinguish which type of tissue

Line 251: Figure 2 Need to correct the unit on the figures as well (E, F, G, H) mg kg-1. Correct units applied in the figures.

Line296: Need to define int he title of table what does the small letter means? . Small letters defined in the table

Line 308: space problem 42%. Space rectified

Line 341: t ha-1 instead of t/ha. Corrections made

Line 393 and 394 enter problem + Line401 and 402 + Line 407 and 408 + Line 544 and 546 (connected to Table 6). Corrections associated with Table 6 made.

Line 410 Need to define this parameter correctly in the part of Materials and Methods chapter. Please add a reference for this parameter. The title of the section now matches the parameter described in the materials and methods. The reference is mentioned in the materials and methods section.

Line 416 Avoid starting the sentence with abbreviation. Phosphorous is better then P, and Line 417 with Nitrogen as well. Abbreviations removed to full words

Line 424 Problem with kg/ha again, see all four figures + try to adjust the Y axe for each other, because different scales are badly influence the reader if it would like to compare them. kgha changed to kg ha-1. The data between the graphs is not meant to be compared. The data within the graphs is important. Hence changing the scale of the Y axis to the same scale would not allow some differences to be viewed particularly for graph B

Line 426 P≤0.05, check all of the figures P<0.05 or that one? For LSD values then we use p≤0.05 where as all other values are based on the F ratio probability.

Line 488 and 496 citation problems. Citation issues fixed in the text .

Yours sincerely David Hall

Reviewer 2 Report

The research presented in the paper concerns various soil management methods to overcome multiple constraints for dryland crops on deep sands. I believe that the subject of the article is very important, and the presented research is of great importance for agricultural practice. The strength of the work is the large number of results that have been thoroughly discussed with the results of other authors. The discussion lacks only a reference to the articles on the influence of the factors being the subject of research on the weed population. In chapter 3.4 the authors describe the results concerning weed infestation and they showed a significant reduction in the number of weeds, so I suggest that they supplement the discussion with this aspect.

I also ask the authors to supplement the methodology with information about which herbicides, insecticides and fungicides (list their active substances) and in what doses were used in the experiment. The authors can add this information to Table 3 or include it in the text of the "Materials and Methods" . Currently, in the methodology, the authors only provide information that the pesticides were used for the entire experiment.

Author Response

Dear Reviewer 2.

Thank you for taking the time to review this paper. Your comments have been very helpful. Please see my responses below which are highlighted in red.

The research presented in the paper concerns various soil management methods to overcome multiple constraints for dryland crops on deep sands. I believe that the subject of the article is very important, and the presented research is of great importance for agricultural practice. The strength of the work is the large number of results that have been thoroughly discussed with the results of other authors. The discussion lacks only a reference to the articles on the influence of the factors being the subject of research on the weed population. In chapter 3.4 the authors describe the results concerning weed infestation and they showed a significant reduction in the number of weeds, so I suggest that they supplement the discussion with this aspect. I have added a discussion relating to weeds and cited findings from other papers.

I also ask the authors to supplement the methodology with information about which herbicides, insecticides and fungicides (list their active substances) and in what doses were used in the experiment. The authors can add this information to Table 3 or include it in the text of the "Materials and Methods" . Currently, in the methodology, the authors only provide information that the pesticides were used for the entire experiment. I am reluctant to add all the herbicides, pesticides and fungicides used in the trial as I believe that it does not add to the understanding of the paper or the trial results. All chemicals were applied to all plots/treatments at the same rates. I have included a detailed catalogue of all the chemicals used in the trial over the 5 years below. Adding all of this information would require a separate table with no comment or discussion around it. I therefore believe that adding this information would detract from the paper’s key findings.

Yours sincerely David Hall

Year

Product

Active ingredient

Rate

Type

2012.00

Hammer

400 g/L carfentrazone-ethyl

45 mL/ha

Herbicide

2012.00

PowerMax

540 g/L Glyphosate

500ml/ha

Herbicide

2012.00

Sakura

850 g/kg Pyroxasulfone

118 g/ha

Herbicide

2012.00

Sprayseed

135 g/Lparaquat 113 g/L Diquat

2 l/ha

Herbicide

2013.00

Sprayseed

135 g/Lparaquat 113 g/L Diquat

2 l/ha

Herbicide

2013.00

Roundup

360 g/L Glyphosate

500ml/ha

Herbicide

2013.00

Altragranz

900g/L Atrazine

450 g/ha

Herbicide

2013.00

Talstar

250 g/L bifenthrin

50 ml/ha

Pesticide

2013.00

Select

240 g/L Clethodim

250 ml/ha

Herbicide

2013.00

Lontrel

600 g/L Clopyralid

150 ml/ha

Herbicide

2014.00

Boxer Gold

800g/l Prosulfocarb 120 g/L Metolachlor

2.5 l/ha

Herbicide

2014.00

Decision

200g/L Diclofop-methyl 20g/L Sethoxydim

1 l/ha

Herbicide

2014.00

Sprayseed

135 g/Lparaquat 113 g/L Diquaot

2 l/ha

Herbicide

2014.00

Roundup

360 g/L Glyphosate

500ml/ha

Herbicide

2015.00

Sprayseed

135 g/Lparaquat 113 g/L Diquat

2 l/ha

Herbicide

2015.00

Atragranz

900g/L Atrazine

1.1 kg/ha

Herbicide

2015.00

Talstar

250 g/L bifenthrin

80 ml/ha

Pesticide

2015.00

Select

240 g/L Clethodim

500 ml/ha

Herbicide

2015.00

Lontrel Granule

600 g/L Clopyralid

100g/ha

Herbicide

2015.00

Atrazine

900g/kg Atrazine

450 g/ha

Herbicide

2015.00

Verdict 520

520 g/l Hayloxyfop

0.1 l/ha

Herbicide

2015.00

Pirimor

500 g/kg Primicarb

0.5 kg/ha

Pesticide

2015.00

Reglone

200 g/l Diquat

3 l/ha

Herbicide

2016.00

Treflan

480 g/L Trifuralin

1 l/ha

Herbicide

2016.00

Sprayseed

135 g/Lparaquat 113 g/L Diquat

2 l/ha

Herbicide

2016.00

Sakura

850 g/kg Pyroxasulfone

118 g/ha

Herbicide

2016.00

Lontrel Granule

600 g/L Clopyralid

100g/ha

Herbicide

2016.00

Flight

35g/l Picolinafen 210g/l Bromoxynil 350 g/LMCPA

540 ml/ha

Herbicide

2016.00

Prosaro

Prothioconazole 210 g/L Tebuconazole 210 g/L

300 ml/ha

Fungicide